# Determinants of Adolescent Reproductive Health in West Java Indonesia: A Cross-Sectional Study

**DOI:** 10.3390/ijerph191911927

**Published:** 2022-09-21

**Authors:** Tetti Solehati, Iqbal Pramukti, Agus Rahmat, Cecep Eli Kosasih

**Affiliations:** 1Department of Maternity, Faculty of Nursing, Universitas Padjadjaran, Bandung 45363, West Java, Indonesia; 2Department of Community Health Nursing, Faculty of Nursing, Universitas Padjadjaran, Bandung 45363, West Java, Indonesia; 3Faculty of Communication Science, Universitas Padjadjaran, Bandung 45363, West Java, Indonesia; 4Department of Emergency and Critical Nursing, Faculty of Nursing, Universitas Padjadjaran, Bandung 45363, West Java, Indonesia

**Keywords:** adolescent reproductive health, attitudes, behavior, demographic

## Abstract

The problem of adolescent reproductive health (ARH) in Indonesia is worrying, especially regarding the Triad Adolescent Reproductive Health (Triad ARH) behavior. Many factors influence ARH behavior. This cross-sectional study explores the association factors between demographic factors, knowledge, and attitudes toward ARH behavior. Six hundred and sixty-eight junior high school and senior high school students were randomly selected from five public schools in Bandung Regency, West Java, Indonesia. The questionnaire used in this study consisted of four parts, namely: (1) demographic data of respondents; (2) knowledge including 20 questions used multiple choice questions; (3) attitude with 12 questions measured using the Likert scale; and (4) behaviors with seven questions measured using Likert scale. Results showed that gender (*p* = 0.006), age (*p* = 0.031), and level of education (*p* = 0.006) were associated with behavior toward ARH behavior, but knowledge (*p* = 0.582), religion (*p* = 0.628), ethnic (*p* = 0.276), and attitude (*p* = 0.094) were not associated with ARH behavior. Multivariate analysis showed that only gender (*p* < 0.010) significantly contributed to ARH behavior. Multivariate analysis showed that gender (OR: 2.168, 95% (CI: 1.204–3.904)) significantly contributed to ARH behavior. Based on the results, it can be concluded that the gender factor influences adolescent reproductive health behavior. This study provides further evidence that to promote positive ARH behavior’s among youth in West Java, Indonesia, gender should be put in place and be sustainable, using the media and the Internet and involving the collaboration of parents, teachers, and peers to improve adolescent reproductive health.

## 1. Introduction

Indonesia is a developing country with a large population of adolescents. Currently, Indonesia is entering the era of demographic bonus, where there is a decrease in the ratio of the non-productive population (age <15 years and ≥65 years) to the productive population (age 15–64 years) [1]. The population of adolescents aged 15–24 is about 17% of the population of Indonesia [2]. Population census data in 2010 showed that the number of adolescents aged 10–24 was quite high, reaching 64 million (27.6%) [3]. Population projections suggest a possible demographic spike by 2030 [4]. The increase in the number of adolescents is likely to create new problems. The age of adolescents with a high curiosity of knowledge will try new things [5]. The Indonesian adolescent population is vulnerable to adolescent health problems [5]. 

Indonesian adolescents are currently experiencing a rapid social change in modern society, where there are changes in their norms, values, and lifestyles [6,7,8]. This situation is at risk of increasing the vulnerability of adolescents to various kinds of problems, one of which is Triad Adolescent Reproductive Health (Triad ARH). Triad ARH forms the three risks faced by adolescents, namely sexuality, HIV/AIDS, and drugs [9]. One of the causes of Triad ARH is inadequate knowledge. The uneven distribution of ARH information in Indonesia causes inadequate information received by adolescents [10]. Knowledge is one of the factors associated with risky behavior in adolescents in Indonesia [11]. Lack of knowledge of adolescents causes adolescents to easily fall into behaviors detrimental to their reproductive health. 

ARH is an important health problem among adolescents in Indonesia [5]. Risky adolescent behavior in Indonesia begins with the consumption of pornography and dating. Adolescents in Indonesia consume a lot of pornography, where pornography consumption can affect sexual behavior [12]. Indonesian adolescents dating at the ages of 15–19 years [13] were 33.3% for girls and 34.5% for boys [14]. Data on teenage pregnancy in Indonesia shows more than 12% of adolescent girls experience unwanted pregnancies and 23% end up having an abortion [2]. In 2015, it was reported that 4.17% of girls and 8.26% of boys had had premarital sex [15]. Premarital sex can increase the risk of contracting sexually transmitted diseases such as HIV/AIDS. According to Menna et al. (2015), worldwide, about 50% of all new cases of HIV occur in adolescents aged 15–24 years [16]. The proportion of HIV infection among adolescents aged 15–24 years in Indonesia continues to increase, namely, 18.4% (in 2014), 19.3% (in 2015), and 21.0% (in 2016) [17]. West Java was one of Indonesia’s five provinces with the highest HIV cases in 2017 [18]. The data show an iceberg phenomenon (only reported). Symptoms of AIDS only appear after 3–10 years of infection. Therefore, most of those who get AIDS have been infected at a younger age. Another problem that threatens children and adolescents is drug abuse. Data from the National Narcotics Agency in 2008 showed that the number of drug users up to 2008 was 115,404, of which 51,986 were aged 16–24 years old, of which there were 5484 school students. Drug cases in Bandung Regency in 2010 recorded 63 cases with 91 suspects [19]. 

Efforts to overcome adolescent reproductive health problems in Indonesia have been promoted since 2003 under Youth Care Health Services (PKPR) [13]. In addition, the National Family Planning Coordinating Board (BKKBN) has also established a risk behavior prevention program for adolescents through an organization called the Youth/Student Information and Counseling Center (PIK R/M) [13]. PKPR and PIK R/M are tasked with training youth to act as peer educators [14]. Unfortunately, the utilization of adolescent reproductive health services in Indonesia tends to be low [13]. This is one of the causes of the inadequate knowledge of adolescents regarding adolescent reproductive health. Poor privacy, confidentiality practices, and provider bias serve as key barriers to care access for adolescent sexual and reproductive health service quality [20]. According to Pradnyani’s, Putra, and Astiti (2019), 1200 adolescents in Bali, Indonesia, showed that some students thought that sexual behavior could be carried out before marriage such as kissing and hugging (48.9%), petting and oral sex (18.7%), and sexual intercourse/vaginal sex (13.8%). A total of 880 out of 1200 teenage students (73.3%) reported having dated someone. Some students reported experiences of petting (14.3%), oral sex (9.8%), vaginal sex (6.5%), and anal sex (2.6%) [21]. Aside from knowledge, attitudes also affect the formation of adolescent behavior. Maulana [22] and Green (2005) [23] explained that there are several factors related to healthy behavior including knowledge, attitude, age, education, ethnicity, religion, and gender. Demographic factors, knowledge, and attitudes allegedly affect adolescent reproductive health behavior [24].

Seeing the problems that arise in adolescents, it is clear that the current protections for adolescents from the dangers of free sex, HIV, and drugs are still not effective enough. An increasing number of adolescents with problems will interfere with the achievement of individual adolescent growth and development tasks (physical growth, mental, emotional and spiritual development), and the task of social growth and the development of adolescents [9]. Various studies have shown that adolescents have very complex problems with their reproductive health, along with the transition period experienced by adolescents [10,25,26] [A study related to reproductive health behavior was carried out by Susanto et al. (2014) in East Java, Indonesia, but was limited to adolescent sexuality and used samples of junior high schools [5], while our study was related to adolescent sexuality, HIV/AIDS, drugs, and samples used in junior high schools and senior high school. Efforts are needed to prevent their reproductive health problems by strengthening education. To determine the right efforts for adolescents, it is necessary to study the demographic data, knowledge, attitudes, and behavior of adolescent reproductive health. Based on the problems that occur regarding adolescent reproductive health in Indonesia, an in-depth analysis is needed to reveal what is going on so that appropriate interventions can be carried out related to adolescent reproductive health problems. The study aimed to analyze the demographic factors, knowledge, and attitudes toward ARH behavior.

## 2. Materials and Methods

### 2.1. Study Design, Setting, and Participants

This study is a quantitative study with a cross-sectional design conducted in several areas in Bandung Regency, West Java, Indonesia, from July 2017 to February 2018. The study population was comprised of all students from three junior high schools (SMP) and two senior high schools (SMA). A total of 12,000 were carried out at the Adolescent Care Health Service (PKPR) and had the Adolescent/Student Information and Counseling Center (PIK-R/M) in Bandung Regency, West Java, Indonesia. The sample used the Slovin formula. Initially there were 670 participants, but there were two participants who did not fill out the answers to the questionnaire completely, so we excluded them from the study, so the final results were 668 students. The sampling technique with conducted with stratified random sampling. Inclusion criteria were active students, willingness to be respondents, and obtaining approval from their parents.

### 2.2. Research Tool

The questionnaire was developed by modifying the WHO questionnaire [27] in terms of knowledge, attitudes, and behaviors related to Triad ARH, and Anggraeni (2018) in terms of knowledge related to Triad ARH [28]. The respondents (self-reported) provided the data independently using a questionnaire that tested for the validity and reliability to 25 students from other junior and senior high schools in the Bandung Regency area. ARH knowledge, attitude, and behaviors obtained Cronbach’s alpha values of 0.87, 0.61, and 0.63, respectively. The validity values were 0.54–0.80, 0.43–0.65, and 0.55–0.82. The questionnaire consisted of four parts, namely: (1) demographic data of respondents which included name, age, gender, religion, ethnicity, level of education; (2) knowledge including 20 questions containing the meaning of ARH, physical characteristics of primary sex changes, male reproductive organs, penile function, understanding of wet dreams in men, impact of masturbation, ways to avoid premarital sex behavior on the reproductive organs women, understanding menstruation, the impact of unwanted pregnancy on adolescents, understanding HIV/AIDS, ways of transmitting HIV/AIDS, preventing HIV/AIDS transmission, body systems being attacked by the HIV virus, understanding drugs, the impact of drug use, how to prevent drug abuse; (3) attitudes with 12 questions containing statements about the attitude of adolescents about sex without marriage bonds by adolescents as an expression of love, is it permissible to have sex without being legally married, everyone is allowed to have premarital sex, sex is part of love that does not need to be limited by the bond of marriage, consultation with peers in overcoming ARH problems, having sex outside is wrong and violates the norm, the importance of maintaining virginity for adolescent girls, sexual relations make couples get to know each other more, parents should not be too closely monitoring the adolescents’ relationships, behave more openly and willing to tell parents, the use of illegal drugs is commonly undertaken by adolescents, the use of illegal drugs is a way to strengthen friendships; and (4) behaviors with seven questions containing behavioral statements about having sex as an expression of sincere love for a boyfriend, having sex before officially getting married, changing partners in sex, maintaining virginity is not important, using illegal drugs when having problems, inviting friends to use illegal drugs, remind friends who use drugs about the dangers of drugs. Knowledge including 20 questions was used in multiple choice questions (four options: one correct and three wrong answers). The wrong answers were assigned ‘0’, while correct answer was assigned a score of ‘1’. The possible scores ranged between 0 and 20. The overall mean score was used for the final analysis of knowledge. The items for attitude and behavior were measured using the 4-point Likert scale. The scale for attitude was: 4 = Strongly Agree; 3 = Agree; 2 = Disagree; and 1 = Strongly Disagree. Negative questions were transformed (‘strongly disagree’ was coded as 4 while ‘strongly agree’ was coded as 1). The scale for behavior was: 4 = Always; 3 = Often; 2 = Sometimes; and 1 = Never. Negative questions were transformed (‘never’ was coded as 4 while ‘Always’ was coded as 1). The possible scores ranged between 12 and 48 for attitude (12 items), and 7–28 for behavior (seven items). The overall mean scores for both attitude and behavior were also used for the final analysis. The higher the participants’ attitude score, the more favorable was their attitude toward ARH. Likewise, the higher the behavior score, the more favorable they were of the ARH.

### 2.3. Procedures

The data collection was carried out in the three junior high schools (SMP) and two senior high schools (SMA) in Bandung Regency, where the questionnaire was self-administered to the students. Each questionnaire lasted 40 min. The researcher briefly introduced the objective of the research and carefully guided the participants in instructions on how to fill out the questionnaire. The questionnaire filling was explained on the first page, and characteristic data on the second page of the questionnaire. Adolescents filled out the questionnaire with the research team and their respective school guardians, having previously obtained the approval of each parent of the respondents and teachers. Adolescents were asked to fill out a questionnaire of knowledge, attitude, and behavior with a choice of four answer items that had been provided. A personal room and seats were arranged to avoid discussions among students during the data collection, and were supervised by the research team and homeroom teachers, so it was not possible to have discussions between fellow students or between students and their teachers. Students who were willing to take part in the research were given merchandise in the form of a stationery kit. Participation was voluntary. The names of all participants were anonymous. The ethical approval from parents was carried out by providing a letter of information and consent to the student’s parents three days before the implementation of the study. The study was carried out the next day after obtaining written consent from the parents. Of the 688 letters sent to the students’ parents, no one refused. Thus, the total was 688 samples.

### 2.4. Statistical Analysis

The data collected was then analyzed using univariate (percentage) analysis, chi-square test, and multiple logistic regression using SPSS software. Univariate analysis in the form of percentages was used to analyze the age, gender, ethnicity, religion, education level, knowledge, and attitudes. Meanwhile, the chi-square analysis was used to analyze the relationship between the demographics, knowledge, attitudes, and adolescent reproductive health behavior. Multivariate analysis was conducted to further find out the relationship demographic factors, knowledge, and attitudes toward ARH behavior. 

### 2.5. Ethics

This research was approved by the Research Ethics Committee of Padjadjaran University (Sertifikat Nomor: 1085/UN6. KEP/EC/2018) and Kesbangpol Bandung Regency. The informed consent form was obtained from the teachers and all parents of the respondents. The respondents could leave the study at any time and decide not to respond to the question. The researcher kept the data confidential.

## 3. Results

The results consisted of the frequency distribution characteristics and the relationship between knowledge, gender, age, religion, ethnicity, level of education, and attitude. The details are shown in Table 1.

A total of 668 participants answered the questionnaire. Table 1 shows that most participants (81.4%) had poor adolescent reproductive health knowledge. Most respondents were female (57.6%), most were aged between 10 and 14 years (50.1%), most followed Islam (96.7%), most were of the Sunda ethnicity (90.1%), and most had the level of education of junior high school (50.3%). It was seen that most respondents (93.75%) were favorable. It also can be seen that gender (*p* = 0.006), age (*p* = 0.031), and level of education (*p* = 0.006) were associated with ARH behavior, but knowledge (*p* = 0.582), religion (*p* = 0.628), ethnicity (*p* = 0.276), and attitude (*p* = 0.094) were not associated with ARH behavior.

Table 2 showed that only gender (*p* < 0.010) significantly contributed to behavior ARH. Multivariate analysis showed that gender (OR: 2.168, 95% (CI: 1.204–3.904)) significantly contributed to behavior ARH. It was reported that gender contributed 2.17 times more to ARH behavior than the others. In contrast, the age and level of education had no significance contributed to the ARH behavior.

## 4. Discussion

In the study, it was found that the majority of the adolescents’ level of knowledge about ARH was categorized as poor. This is in line with the research of Utmo et al. (2014) on adolescents in Indonesia and Yazdi et al. on adolescents in Iran [29,30]. This problem is probably due to inadequate ARH information obtained by adolescents due to the uneven distribution of information related to ARH in Indonesia [10]. Many adolescents experience barriers to accessing reproductive health information and care [31]. Indonesian youth also feel ashamed and taboo to discuss the topic of sexuality and ARH with parents and teachers [32,33]. Apart from that, the parents themselves believe that adolescents are too young to understand and that discussing reproductive health with adolescents will encourage sexual experimentation [32]. Providing sexual education to unmarried youth is culturally unacceptable in most Muslim societies [34] including Indonesia. Parents, teachers, and policy-makers believe that it can result in premature sexual activity [35]. This causes adolescents to seek their own information through mass media and their peers [36] and is considered the main source of information [37]. However, this information does not necessarily provide adequate information regarding ARH. Ideally, health workers in the community including in schools should provide health promotion about ARH to adolescents and carry out ongoing evaluations.

The results showed that the average knowledge of adolescents was poor. However, the results showed that their attitudes and behavior supported ARH. This is probably due to the role of culture. In this study, the majority of respondents were Sundanese. Sundanese culture has a taboo culture such as prohibitions from doing unusual things, one of which is free sex and the use of illegal drugs. Their obedience to their ancestors guarantees their safety in this world and the hereafter [38]. Culture can be a support for adolescent behavior. Adolescents study health behavior in the socio-cultural context of society, family, and peers [39]. Asian culture disapproves of sex outside of marriage [40] and the value of virginity is taken seriously. A girl who was still a virgin when she got married brought great pride to her family [41]. Moreover, this is supported by the role of religion, where the majority of respondents in this study were Muslim. Religion has a role in adolescents. Islam strictly forbids its followers from committing sins such as having free sex, having sex before marriage, and using drugs, so that they will avoid HIV/AIDS. In Muslim countries such as Iran and most other Muslim countries, refusing sex outside of marriage is an important barrier to the fight against HIV/AIDS [34]. In Muslim countries, the chastity of girls is an honor for their families [42,43].

In further analysis, it was found that the gender variable affected ARH behavior. Cultural and religious influences caused parents to be tough and instill fear in their adolescent, a fear of the dangers of negative sexual and reproductive health [41], especially in adolescent girls. ARH behavior develops according to gender and norms [5]. Gender has an important role in ARH [44]. According to Odimegwu and Somefun (2017), gender roles can encourage or prevent risky behavior [45]. The majority of respondents in this study were women. According to Susanto et al. (2016), the prevalence of active reproductive health behaviors is higher in boys than in girls [5]. Girls are more obedient to their parents’ commands and prohibitions than boys. According to Wenxin et al. (2006), girls are more oriented toward family relationships and obedience, while boys are oriented toward autonomy and independence [46]. Indonesian youth need educational programs that can improve the knowledge, attitudes, and behavior, and encourage the formation of positive attitudes toward norms and gender related to ARH [5].

Age, level of education, knowledge, and attitude in this study did not influence adolescent behavior. The results of this study contradict Maulana (2009), Green (2005), and Mmari (2009) [22], who argued that age, level of education, knowledge, and attitude can affect a person’s behavior [23,24]. Increasing age will increase one’s knowledge. With the increasing level of knowledge, one can distinguish between good and bad things. Good knowledge will affect one’s attitude in supporting good things and vice versa. In this study, it was also found that religion and ethnicity did not affect adolescent behavior. The results of the study contradict Motsomi et al. (2016), who stated that culture and religion have a role in behavior [32], and Odimegwu and Somefun (2017), who stated that ethnicity can encourage or prevent risky behavior [45]. Culture, race, and religion influence ARH [5]. Religion and culture can reduce the level of sexual activity among adolescents [47]. These factors do not affect adolescent behavior because adolescents in this study were more dominated by the characteristics of the obedient nature of adolescents on the recommendations and prohibitions of parents or teachers who were figures they trusted. Normatively, parents regulate the moral and prudence of their adolescents [48]. In addition, Indonesia has legal regulations regarding Triad ARH. Indonesia is the most prominent Islamic country with a legal prohibition on pornography [5].

Humans are unique creatures including adolescents. An adolescent is defined as an individual between the ages of 10–19 years [49,50]. It is necessary to strengthen behavior that supports ARH on an ongoing basis so that they do not fall into the wrong ARH. The health care professionals and educators need to be involved in the ARH education promotion program [51]. Nurses are one of the professions that have an essential role in adolescent health care including ARH. As the main focus of adolescent nursing care, caring is needed as the primary mechanism for effective health promotion [52]. Nurses must advocate for the legitimacy and importance of treatment modalities in promoting adolescent health [52] including ARH. Thus, adolescents are expected to perform ARH well, which can indirectly improve their well-being.

## 5. Limitations

This study had several potential limitations to our results in that the sample size used was limited to a few senior high schools and junior high schools in Bandung district. There was less variety in the religious samples. It is important to develop further research in a wider range of senior high and junior high schools, and with a variety of religions.

## 6. Conclusions

Based on the study results, it can be concluded that most knowledge can be categorized as poor, but has attitudes and behaviors that support positive ARH. One main factor influencing adolescent behavior in doing their ARH is gender. Thus, an intervention method is needed to improve adolescent behavior related to positive ARH by taking into account their gender, which can be carried out sustainably using media and the Internet and involves the collaboration of parents, teachers, and peers to increase positive ARH in improving adolescent well-being.

## Figures and Tables

**Table 1 ijerph-19-11927-t001:** The respondents’ characteristics and the bivariate analysis of adolescent reproductive health in Bandung Regency (*n* = 668).

Characteristics	f	%	Behavior		*χ* ^2^	*p*
			Favorable	Unfavorable		
Knowledge						
Good	124	18.6	501	43	302	0.582
Poor	544	81.4	116	8		
Gender						
Female	385	57.6	365	20	7.672	0.006
Male	283	42.4	252	31		
Age						
10–14 years	335	50.1	302	33	4.680	0.031
15–19 years	333	49.9	315	18		
Religion						
Islam	646	96.7	598	48	1.742	0.628
Christian	16	2.4	14	2		
Buddhist	5	0.7	4	1		
Hindu	1	0.1	1	0		
Ethnicity						
Sunda	602	90.1	558	44	2.577	0.276
Java	41	6.1	38	3		
Other	25	3.7	21	4		
Level of education						
Junior high school	336	50.3	301	35	7.419	0.006
Senior high school	332	49.7	316	16		
Attitude						
Favorable	626	93.75	581	45	2.811	0.094
Unfavorable	42	6.30	36	6		

**Table 2 ijerph-19-11927-t002:** The multivariate result logistic regression showing the odds ratio between the socio-demographics and ARH behavior.

Variables	OR	95% CI		*p*
		Lower	Upper	
Gender	2.168	1.204	3.904	0.010
Age	1.553	0.450	5.360	0.487
Level of education	0.305	0.086	1.083	0.066

## Data Availability

The datasets generated and analyzed of this article are available from the corresponding author.

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
