# Peer review of "Determinants of Adolescent Reproductive Health in West Java Indonesia: A Cross-Sectional Study"

_ijerph, 2022, doi:10.3390/ijerph191911927_

Round 1
Reviewer 1 Report
The research has a high level of importance because it deals with the health of adolescents. I believe the main audience is Asian, because the local data (Indonesia), due most likely to religious and cultural issues, differs from most of the Eastern region and some of the Western region.
The study was carried out between 2017-2018, 4 to 5 years ago, this can generate outdated and modified data due to the pandemic, however it does not imply its importance. The number of samples is consistent. I suggest the inclusion of the form used with the questions attached. I felt a lack of data on teenage pregnancy that is relevant to the junction of the themes. There is no mention of condom use or knowledge by these adolescents, as it would be a way of preventing HIV and other sexually transmitted diseases. The design is appropriate. The conclusion is very clear and coherent, the lack of clarification promotes the growth of reproductive diseases.
Author Response
Dear Reviewer,
I have corrected my manuscript as you suggested. There is an additional reference number 48, namely Cumsille which replaces Montalti (originally number 48), there is an additional reference number 49, namely the Ministry of Health which shifts Saewyc (originally number 49), there is an additional reference number 50, namely WHO. Montalti's reference to be 51 and Saewyc's reference to be 52. Thank you very much
Warm Regards
Tetti Solehati

Reviewer 2 Report
This is an interesting paper and examines aspects related to reproductive health, but actually not defined what outcome is actually explored.
See my comments:
1. The title is somewhat misleading since you mention 'adolescents', but include children who are neither teenagers nor adolescents.
2. Please define what you mean by 'behaviour towards ARH'. It can mean positive or negative behaviour or a certain view towards ARH. This should be very clear, otherwise, the abstract does not explain what you are reporting. Also in the abstract, you do not mention what methods are used (no mention of multiple choice questionnaires). And in the method section, it should be mentioned if the questionnaires are validated tools or questions you thought may be interesting. This has implications for the validity of the results.
3. TRIAD is written in capitalized letters. Is it an abbreviation or just denotes a triad (use small letters)? In the abstract it is not clear what triad ARH actually means.
4. 100% participation rate is very rare. Was the element of free choice real? Also please discuss bias and non-honest answers if the questionnaires were filled out together with a grown-up. Later, in the Discussion, you mention how discussing sexuality with parents and teachers is for many embarrassing and taboo. How do children fill out questionnaires on the topic with a guardian beside them helping them out? Also, you mention parents and religion find sexual education unacceptable. How do you explain that 100% of all parents agreed to let their children participate in a study on reproductive health?
5. You write: 'Increasing numbers of adolescents with problems'. What problems do you mean? Do you mean attitudes towards reproductive health? And if 'yes' what are the problems? Problems could otherwise mean anything.
6. Results: You write 'the majority of respondents (93,75 %) were favorable.' What do you mean by this? Actually, it is not well explained what you are measuring as favourable.
7. Table 2: Please explain what is measured in the Title of the table. And in the text what variable is the independent one? p=0.066 - you call this a significant difference. p<0.05 will usually be significant, but 0.07 is higher. And the OR of 1.3 with CI: 0.086-1.083 can not be assigned significance. A CI of 0.9-1.1 actually denotes that the true risk may be either higher or lower, i.e. does not explain anything.
8. You mention that religion and ethnicity did not affect adolescent behaviour. Having in mind that almost all were of the same religion (close to 97%). There are actually not enough participants in the other groups to be able to calculate any significance even if it were so. Therefore I do not think this finding contradicts other studies, it only shows that the low numbers in the other religions-groups were not suitable for any conclusion on this.
9. In the discussion and conclusions you use ARH, but do not define if you mean good ARH, bad ARH, positive, etc. Without an adjective 'health' is not explained well. Also, you conclude with: adolescents doing their health (AHR). Doing health does not mean anything. Also, are 10-12-year-olds considered adolescents? Please explain better.
Minor:
Add a reference (line 252).
Abbreviations should be explained (NAFZA?)
Typos: There are several typos, both in the abstract and in the paper. Please read through.
Author Response

(The authors gave the same response as above.)

Round 2
Reviewer 2 Report
The authors have adequately addressed all my comments and changed in the text.
Some small typos still need to be corrected